# Comparison of Methicillin-Resistant *Staphylococcus aureus* Isolates from Cellulitis and from Osteomyelitis in a Taiwan Hospital, 2016–2018

**DOI:** 10.3390/jcm8060816

**Published:** 2019-06-07

**Authors:** Kuo-Ti Peng, Tsung-Yu Huang, Yao-Chang Chiang, Yu-Yi Hsu, Fang-Yi Chuang, Chiang-Wen Lee, Pey-Jium Chang

**Affiliations:** 1Department of Orthopaedic Surgery, Chang Gung Memorial Hospital, Chiayi 61363, Taiwan; mr3497@cgmh.org.tw (K.-T.P.); yuyi1158@cgmh.org.tw (Y.-Y.H.); 2College of Medicine, Chang Gung University, Taoyuan 33303, Taiwan; 3Division of Infectious Diseases, Department of Internal Medicine, Chang Gung Memorial Hospital, Chiayi 61363, Taiwan; r12045@adm.cgmh.org.tw; 4Graduate Institute of Clinical Medical Sciences, College of Medicine, Chang Gung University, Taoyuan 33303, Taiwan; 5Department of Nursing, Chang Gung University of Science and Technology, Chiayi 61363, Taiwan; yaochang.chiang@gmail.com (Y.-C.C.); cwlee@gw.cgust.edu.tw (C.-W.L.); 6Division of Basic Medical Sciences, and Chronic Diseases and Health Promotion Research Center, Chang Gung University of Science and Technology, Chiayi 61363, Taiwan; 7Department of Laboratory Medicine, Chang Gung Memorial Hospital, Chiayi 61363, Taiwan; daisy5670@cgmh.org.tw; 8Research Center for Industry of Human Ecology and Research Center for Chinese Herbal Medicine, Chang Gung University of Science and Technology, Taoyuan 33303, Taiwan; 9Department of Rehabilitation, Chang Gung Memorial Hospital, Chiayi 61363, Taiwan; 10Department of Nephrology, Chang Gung Memorial Hospital, Chiayi 61363, Taiwan

**Keywords:** MRSA, cellulitis, osteomyelitis, periprosthetic joint infections, antibiotic resistance

## Abstract

Methicillin-resistant *Staphylococcus aureus* (MRSA) causes superficial infections such as cellulitis or invasive infections such as osteomyelitis; however, differences in MRSA isolates from cellulitis (CL-MRSA) and from osteomyelitis (OM-MRSA) at the same local area remain largely unknown. A total of 221 MRSA isolates including 106 CL-MRSA strains and 115 OM-MRSA strains were collected at Chang-Gung Memorial Hospital in Taiwan between 2016 and 2018, and their genotypic and phenotypic characteristics were compared. We found that OM-MRSA isolates significantly exhibited higher rates of resistance to multiple antibiotics than CL-MRSA isolates. Genotypically, OM-MRSA isolates had higher proportions of the SCC*mec* type III, the sequence type ST239, and the *spa* type t037 than CL-MRSA isolates. Besides the multidrug-resistant lineage ST239-t037-SCC*mec*III more prevalent in OM-MRSA, higher antibiotic resistance rates were also observed in several other prevalent lineages in OM-MRSA as compared to the same lineages in CL-MRSA. Furthermore, when prosthetic joint infection (PJI) associated and non-PJI-associated MRSA strains in osteomyelitis were compared, no significant differences were observed in antibiotic resistance rates between the two groups, albeit more diverse genotypes were found in non-PJI-associated MRSA. Our findings therefore suggest that deep infections may allow MRSA to evade antibiotic attack and facilitate the convergent evolution and selection of multidrug-resistant lineages.

## 1. Introduction

Methicillin-resistant *Staphylococcus aureus* (MRSA) is one of the most important pathogens causing both healthcare- and community-associated infections [1]. MRSA can cause a wide variety of clinical diseases, including skin and soft tissue infections (SSTIs), sepsis, pneumonia, meningitis, infective endocarditis, bacteremia, periprosthetic joint infections (PJIs), and osteomyelitis [2,3]. Previous studies have shown that in addition to the resistance to β-lactam antibiotics, clinical MRSA isolates have increasingly acquired resistance to many non-β-lactam antibiotics. Particularly, MRSA strains with reduced susceptibility to vancomycin, a last line of defense against MRSA, have now been frequently reported in different countries [4,5,6].

MRSA infections are generally classified by two major categories based on epidemiologic definition: healthcare-associated MRSA (HA-MRSA) infection and community-associated MRSA (CA-MRSA) infection where patients are divided by the presence or absence of specific risk factors for healthcare exposure, respectively [7,8]. Several previous studies have revealed that HA-MRSA isolates mainly carry staphylococcal cassette chromosome *mec* (SCC*mec*) types I, II or III, and are typically resistant to multiple non-β-lactam antibiotics, whereas CA-MRSA isolates harbor SCC*mec* types IV or V, and are susceptible to non-β-lactam antibiotics [9,10,11,12,13,14]. Intriguingly, a growing body of evidence shows that many CA-MRSA isolates have emerged as the dominant clones in East Asia, including Taiwan [15,16,17,18]. Besides the SCC*mec* typing, different DNA sequence-based approaches such as multilocus sequence typing (MLST) and *spa* typing allow for the detection of closely related MRSA strains [19,20,21,22]. The presence of the toxin gene encoding Panton–Valentine leukocidin (PVL) can also be another genetic feature of CA-MRSA [23,24,25,26]. According to the previously reported studies, HA-MRSA may theoretically have a different impact on patient outcomes in comparison with CA-MRSA. However, due to the fact that CA-MRSA enters the healthcare system and HA-MRSA spreads into the community, the traditional distinction between HA-MRSA and CA-MRSA now tends to be blurry and has little predictive value for clinical practice [27,28]. Although genotyping and antibiotic susceptibility pattern can also be used for classifying MRSA infections as HA-associated or CA-associated, each of the approaches has distinct advantages and disadvantages [28,29]. 

Due to the regional differences in antibiotic usage, the epidemiological transition of MRSA strains may vary widely in different geographical areas. Although considerable variation in the proportion of specific MRSA genotypes could be expected in different geographical areas, it remains unclear whether MRSA strains from superficial infections (such as SSTIs of the skin; cellulitis) and from invasive infections (such as osteomyelitis) in the same area exhibit similar clonal distributions and similar phenotypic characteristics. In the study, we aimed to compare the genotypic and phenotypic differences of MRSA strains recovered from cellulitis (superficial infection) and from osteomyelitis (invasive infection) at a 1300-bed tertiary teaching hospital in southern Taiwan.

## 2. Methods

### 2.1. Patients and MRSA Isolates

Clinical MRSA isolates were collected from Chang Gung Memorial Hospital, Chiayi, Taiwan during the period between 2016 and 2018. This study was approved by the Institutional Review Board of Chang Gung Memorial Hospital (Number: 201600459B1). Informed consent was obtained from all participants. A total of 221 MRSA isolates, including 106 strains from cellulitis (CL-MRSA) and 115 strains from osteomyelitis (OM-MRSA), were obtained from different clinical specimens such as cutaneous abscess, wound secretion or tissues including skin, subcutaneous soft tissue or bone. Duplicate cultures from the same patient were excluded from the study. Medical record review, using standardized clinical criteria and isolation of MRSA from a normally sterile site, was used to define and classify infections. All MRSA isolates were recognized by coagulase testing and by their resistance to oxacillin according to the Clinical and Laboratory Standards Institute (CLSI) guidelines [30]. Clinical and demographic data of patients were collected from the electronic medical records. 

### 2.2. Antibiotic Susceptibility Test

The minimum inhibitory concentration (MIC) of ciprofloxacin, fusidic acid, gentamicin, rifampicin, trimethoprim-sulfamethoxazole (TMP-SMX), and vancomycin was determined with the E-test method. Briefly, MRSA isolates were grown in tryptic soy both (3 mL) at 37 °C overnight, and then bacterial suspension was prepared with the turbidity equivalent to a 0.5 McFarland standard (1 to 2 × 10^8^ CFU/mL). The E-test gradient strip (Etest^®^, bioMérieux SA, Marcy-l’Étoile, France) was placed on an inoculated agar surface. The MIC value was read from the scale in terms of μg/mL where the pointed end of the ellipse intersected the strip. The threshold of MIC results was chosen based on the guidelines of Clinical Laboratory Standards Institute (CLSI) or previous studies [30,31].

### 2.3. SCCmec Typing

The SCC*mec* types of MRSA isolates were determined by multiplex PCR. Genomic DNA of MRSA strains was extracted using the Mini gDNA bacteria Kit (Geneaid, Taipei, Taiwan) coupled with the additional use of lysostaphin to lyse bacteria. The PCR primer sets and amplification conditions used in the study have been described previously [32]. PCR products (5 µL) were analyzed on 1.5% agarose gels, and the corresponding SCC*mec* type was determined according to the band pattern described previously [32,33].

### 2.4. Multilocus Sequence Typing (MLST)

Seven gene loci including carbamate kinase (*arcC*), shikimate dehydrogenase (*aroE*), glycerol kinase (*glp*), guanylate kinase (*gmk*), phosphate acetyltransferase (*pta*), triosephosphate isomerase (*tpi*), and acetyl coenzyme A acetyltransferase (*yqiL*) were used in the genotyping. Primers used in the experiments have been described previously [19]. The allelic profiles or sequences types (STs) were determined by the *S. aureus* MLST database (http://saureus.mlst.net/).

### 2.5. spa Typing

The X region of the *spa* gene of each MRSA isolate was amplified by PCR and the amplified products were sequenced. Variation of short repeated regions in the *spa* gene was analyzed through Ridom StaphType software (Ridom GmbH, Würzburg, Germany) [20].

### 2.6. Detection of Panton–Valentine Leukocidin (pvl) Gene

The Panton–Valentine leukocidin (PVL) gene was detected using the primer set, Luk-PV-1 (ATCATTAGGTAAAATGTCTGGACATGATCCA) and Luk-PV-2 (GCATCAAGTGTATTGGATAGCAAAAGC), which resulted in a 433-bp gene product [34]. 

### 2.7. Statistical Analysis

Data collection and statistical analyses were performed using GraphPad Prism version 7.0 (GraphPad Software Inc., San Diego, CA, USA). The student t test, ANOVA and Chi-square test were used to analyze the differences between groups. *p* value < 0.05 was considered statistically significant.

## 3. Results

### 3.1. Clinical Characteristics

There were 106 cellulitis MRSA (CL-MRSA) isolates and 115 osteomyelitis MRSA (OM-MRSA) isolates included in the study. From medical records, approximately 65% of MRSA strains were obtained from male patients (Table 1). The average age of patients with cellulitis was younger than that of patients with osteomyelitis (54 ± 21 vs. 60 ± 18; *p* < 0.05). OM-MRSA strains were predominantly isolated from limbs (92%), whereas CL-MRSA strains were isolated across the entire body such as head and neck (28%), trunk (25%), and limbs (46%). Furthermore, analysis of the patient characteristics such as underlying diseases or medical conditions revealed that OM-MRSA infection occurred more frequently than CL-MRSA infection in patients with chronic hepatitis and cirrhosis (CHC) or chronic kidney disease (CKD), and in patients that had experienced hospitalization or surgery in the past 6 months (Table 1).

### 3.2. Antibiotic Resistance Profiles of CL-MRSA and OM-MRSA Isolates

The antibiotic resistance patterns of the isolated CL-MRSA and OM-MRSA strains examined by MIC analysis and susceptibility testing are shown in Table 2. As compared to CL-MRSA isolates, we found that OM-MRSA isolates significantly exhibited higher resistance rates to ciprofloxacin (60% vs. 35%), fusidic acid (22% vs. 3%), gentamicin (51% vs. 23%) and trimethoprim-sulfamethoxazole (TMP-SMX) (26% vs. 7%). Although all CL-MRSA and OM-MRSA strains were typically susceptible to rifampicin and vancomycin, a limited number of OM-MRSA strains appeared to be less susceptible to rifampicin (MIC ≥ 4 µg/mL; *n* = 2) and vancomycin (MIC ≥ 2 µg/mL; *n* = 6).

### 3.3. Genotypic Characterization of CL-MRSA and OM-MRSA Isolates

Molecular genotyping analyses of CL-MRSA and OM-MRSA isolates are shown in Table 3. Analysis of the SCC*mec* typing revealed that SCC*mec* type IV was the most prevalent for both CL-MRSA and OM-MRSA isolates (56% and 48%, respectively). However, we noticed that there were marked differences in the proportions of SCC*mec* types III and V between CL-MRSA and OM-MRSA (6% vs. 24% for SCC*mec* type III; 39% vs. 26% for SCC*mec* type V). MLST analysis showed that 11 and 13 distinct STs were found in CL-MRSA and OM-MRSA isolates, respectively. ST59 was the most predominant sequence type for both CL-MRSA (51%) and for OM-MRSA (32%). In addition to ST59, the other remaining STs for CL-MRSA were ST8 (21%), ST30 (8%), ST239 (7%) and ST45 (6%), whereas the other remaining STs for OM-MRSA were ST239 (24%), ST8 (21%), ST45 (6%) and ST5 (3%). Noteworthily, ST239 was found at a much higher proportion in OM-MRSA than in CL-MRSA. By analysis of the polymorphic X region of *spa* gene, 16 *spa* types for CL-MRSA and 19 *spa* types for OM-MRSA were identified. For CL-MRSA, the most prevalent *spa* type was t437 (36%), followed by t008 (18%), t441 (11%), t019 (8%) and t037 (7%). In contrast, the most prevalent *spa* type for OM-MRSA was t037 (25%), followed by t437 (23%), t008 (17%), t441 (8%) and t1081 (5%). Herein, there were three isolates characterized with new *spa* types found in CL-MRSA isolates (Appendix A). Additionally, when the frequency of PVL toxin genes was compared between CL-MRSA and OM-MRSA, we found that 65% of CL-MRSA isolates were positive for PVL genes, while 56% of OM-MRSA isolates were positive for PVL genes.

### 3.4. Association between Antibiotic Resistance and Genotypes in CL-MRSA and OM-MRSA

The relationships between the phenotypic antibiotic resistance and MLST genotypes for CL-MRSA and OM-MRSA were first examined and shown in Table 4 and Table 5, respectively. Although different sequence types in CL-MRSA or in OM-MRSA might confer heterogeneous antibiotic resistance patterns, we here found that ST8 and ST239 consistently showed high rates of resistance to ciprofloxacin (up to 80%) in CL-MRSA and in OM-MRSA. Notably, ST239 in both CL-MRSA and OM-MRSA also exhibited high resistance to gentamycin and TMP-SMX (>85%). On the other hand, as compared to the same STs in CL-MRSA, we found that four predominant STs in OM-MRSA, including ST8, ST45, ST59 and ST239, appeared to have higher resistance rates to fusidic acid. Similarly, the higher resistance rate to gentamicin was also observed in ST8, ST45 and ST59 of OM-MRSA as compared to those of CL-MRSA. To further investigate the association between MRSA genotypes and antibiotic resistance profiles for CL-MRSA and OM-MRSA, a combined genotyping analysis that includes SCC*mec*, MLST and *spa* typing was conducted (Table 6 and Table 7). We showed that several prevalent MRSA lineages including ST8-t008-SCC*mec*IV (*n* = 18), ST45-t1081-SCC*mec*IV(V) (*n* = 6), ST59-t437-SCC*mec*IV (*n* = 15), ST59-t437-SCC*mec*V (*n* = 12), and ST59-t441-SCC*mec*V (*n* = 6) in OM-MRSA had higher rates of resistance to fusidic acid and gentamycin than the same lineages in CL-MRSA.

### 3.5. Correlation between MRSA Genotypes and Diseased States of Patients with Cellulitis and Osteomyelitis

Medical record review and analysis revealed that patients with OM-MRSA generally underwent more surgeries and more hospitalization than patients with CL-MRSA (Table 8). When patients with different STs (including ST8, ST59 and ST239) in cellulitis were compared, we found that infection with ST239 substantially required more surgeries for a full recovery than infection with ST8 or ST59. These results suggested that ST239 infections caused more refractory cases than ST8 or ST59 infections in cellulitis. In contrast, we did not find significant differences in numbers of operations between ST239 and ST8 or ST59 infections in osteomyelitis, raising the possibility that ST8 and ST59 MRSA in osteomyelitis might have acquired additional abilities to escape from standard treatments (including antibiotic treatment)

Specific health conditions such as chronic liver or kidney diseases could be linked to increased morbidity of osteomyelitis caused by different MRSA genotypes. After review of patient’s clinical medical records including the frequencies of surgical operations and hospital admissions, and time intervals between operations or between admissions, we found that patients with chronic liver/kidney diseases were critically associated with enhanced morbidity of osteomyelitis caused by ST239, but not by ST8 or ST59 (Table 9).

### 3.6. Differences in Antibiotic Susceptibility and Genotypes Between PJI- and Non-PJI-Associated OM-MRSA

Among 115 OM-MRSA isolates, there were 26 isolates (23%) obtained from PJIs, designated as “OM-MRSA(PJI)”, and 89 isolates (77%) from non-PJIs, designated as “OM-MRSA(non-PJI)”. When the isolated OM-MRSA(PJI) and OM-MRSA(non-PJI) strains were compared, we found few differences in their antibiotic resistance profiles (Table 10). Intriguingly, genotyping analysis revealed that the distributions of SCC*mec* types, ST types, and *spa* types were considerably different (Table 11). In addition to the differential proportions of the SCC*mec* types III and IV in these two groups, we also found that OM-MRSA(non-PJI) isolates had more diverse ST subtypes (12 vs. 6) or spa subtypes (18 vs. 7) than OM-MRSA(PJI) strains. In the OM-MRSA(PJI) group, the genotype ST239-t037-SCC*mec*III was the most prevalent lineage (*n* = 9; 35%), followed by ST59-t437-SCC*mec*V (*n* = 5; 19%), ST59-t437-SCC*mec*IV (*n* = 3; 12%), and ST8-t008-SCC*mec*IV (*n* = 3; 12%) (Appendix A). In the OM-MRSA(non-PJI) group, the genotype ST239-t037-SCC*mec*III was the most prevalent lineage (*n* = 16; 18%), followed by ST8-t008-SCC*mec*IV (*n* = 15; 17%), ST59-t437-SCC*mec*IV (*n* = 12; 14%), and ST59-t437-SCC*mec*V (*n* = 7; 8%) (Appendix A). When the most prevalent sequence types including ST8, ST59 and ST239 in these two groups were compared, their antibiotic resistance patterns were basically similar between PJIs and non-PJIs in osteomyelitis (Appendix A). These results implicated that convergent evolutions of different MRSA clones were driven in MRSA osteomyelitis including PJIs and non-PJIs.

## 4. Discussion

MRSA has become a serious public health problem today, and numerous MRSA clones have now spread globally since the first MRSA strain was reported in 1961 [35,36]. Although many previous studies have often focused on the distinction between HA-MRSA and CA-MRSA based on epidemiologic, genetic or microbiological profiles, the classification between HA-MRSA and CA-MRSA is now becoming more difficult and has little predictive value for clinical practice because HA-MRSA penetrates into the community and CA-MRSA circulates in the healthcare system. MRSA could lead to both superficial tissue infections (such as skin and muscle) and invasive tissue infections (such as bone and bloodstream). Deep tissue infection by MRSA is generally considered to be more difficult to treat than superficial tissue infection by MRSA. Herein, we characterized and compared the antibiotic resistance profiles and genotypes of MRSA isolates from cellulitis (CL-MRSA) and from osteomyelitis (OM-MRSA) at a regional hospital in Taiwan. We indeed found that OM-MRSA strains are more resistant to many antibiotics (including ciprofloxacin, fusidic acid, gentamicin, rifampicin, TMP-SMX and vancomycin) than CL-MRSA strains. 

In our genotyping analysis, two major conclusions could be drawn to explain differential antibiotic resistance phenotypes between CL-MRSA and OM-MRSA. First, a higher proportion of the ST239-t037-SCC*mec*III lineage was seen in OM-MRSA than in CL-MRSA. The ST239-t037-SCC*mec*III lineage is one of the most successful HA-MRSA disseminated globally, which is typically resistant to many non-β-lactam antibiotics. Second, several other prevalent lineages in OM-MRSA, such as ST8-t008-SCC*mec*IV, ST45-t1081-SCC*mec*IV(V), ST59-t437-SCC*mec*IV, ST59-t437-SCC*mec*V and ST59-t441-SCC*mec*V, have become more resistant to fusidic acid and gentamycin as compared to the same lineages in CL-MRSA. Noteworthily, ST59 with SCC*mec*IV or SCC*mec*V is now the most common CA-MRSA lineage in Taiwan. On the other hand, we did not find the association between antibiotic resistance patterns and PVL status in either CL-MRSA or OM-MRSA in the study. 

To link the association of disease severity with MRSA subtypes, we reviewed patient’s clinical medical records. Generally, patients with OM-MRSA caused more therapy-refractory cases than patients with CL-MRSA (Table 8, numbers of surgical operations). During the course of the study, we noticed that patients with ST239 infection caused more severe illness than patients with ST8 or ST59 infection in the cellulitis group. However, there were no significant differences in disease severity between ST239 and ST8 or ST59 infections in the osteomyelitis group. These findings raise a possibility that ST8 and ST59 MRSA in osteomyelitis may have already acquired some additional virulence-associated factors to allow them escape from standard treatments.

Due to the unique mode of MRSA transmission in PJIs, we additionally compared the potential differences in antibiotic susceptibility and genotypes between PJI- and non-PJI-associated OM-MRSA isolates. We found that OM-MRSA(non-PJI) was composed of more-diverse clonal lineages (especially lineages from the community origin) than OM-MRSA(PJI), indicating that only a limited range of MRSA strains got into PJI in the osteomyelitis group. Despite the high diversity of MRSA subtypes present in OM-MRSA(non-PJI) isolates, there were no significant differences in antibiotic resistance profiles between OM-MRSA(PJI) and OM-MRSA(non-PJI) isolates (Table 10). These results implicate that convergent evolution might have occurred in the isolated OM-MRSA(PJI) and OM-MRSA(non-PJI) strains in response to the similar selective pressures. In the osteomyelitis group, we additionally observed that kidney/liver diseases are critically associated with worse outcomes for patients with ST239 infections; however, the comorbidity did not further affect the outcomes of patients with ST59 or ST8 infections (Table 9). Due to many unique characteristics of ST239 infections in cellulitis or in osteomyelitis, it would be therefore important to pay more attention for the treatment of patients with the classical HA-MRSA infection if the pathogenic genotype is known.

To achieve optimal clinical outcomes and prevent further spread of highly virulent human-adapted MRSA lineages, the correct use of antibiotic therapy is vitally important and necessary. For treating osteomyelitis MRSA infections, vancomycin is considered as an effective treatment [37]. However, the incidence of MRSA isolates that are resistant to vancomycin, such as VISA (vancomycin-intermediate *S. aureus*) and VRSA (vancomycin-resistant *S. aureus*), appears to be increasing worldwide. In the study, there were six VISA isolates found in osteomyelitis samples. Previous studies have suggested that TMP-SMX is a recommended therapy when MRSA infections received a vancomycin MIC of ≥2 µg/mL [31]. High-dose TMP-SMX combined with surgical debridement were effective for treating PJI-induced chronic osteomyelitis [38]. Based on our current studies, these suggestions are generally acceptable for osteomyelitis caused by ST59 or ST8 infection; however, there were not applicable for osteomyelitis caused by ST239 or the closely related ST241 infections (Table 5). Furthermore, ciprofloxacin, one of the most commonly used fluoroquinolone antibiotics, could be suitable for treating ST59-infected cellulitis and osteomyelitis, but not for ST8- and ST239-induced cellulitis and osteomyelitis (Table 5 and Table 6). Since ciprofloxacin has been reported to induce liver injury [39], its clinical use in osteomyelitis is still limited. Fusidic acid and rifampicin are currently used for treating MRSA osteomyelitis and PJIs [37,38,40]. Based on our results, adjunctive rifampin therapy may be able to improve outcomes from most MRSA infections. Currently, prevention of the bacterial resistance to antibiotics still remains a challenge. Although the molecular mechanisms regarding the emergence of these multi-drug resistant MRSA lineages in the study still remain unknown, it is possible that these successfully adapted MRSA lineages may have acquired antibiotic resistance by chromosomal mutations, and/or by obtaining an antibiotic resistance gene from another bacterium via mobile genetic elements including phages and plasmids [41,42].

In conclusion, we have shown that OM-MRSA isolates have higher antibiotic resistance rates than CL-MRSA isolates, and the higher antibiotic resistance rates for OM-MRSA may be attributed to a higher prevalence of the ST239-t037-SCC*mec*III lineage, and a greater evolution of several common MRSA lineages that acquire antibiotic resistance. Understanding the genotypic and phenotypic characteristics of MRSA infections in superficial and deep tissues is important for ongoing surveillance of multidrug-resistant MRSA lineages.

## Figures and Tables

**Table 1 jcm-08-00816-t001:** Basic demographics of patients with MRSA infections.

Characteristic	Cellulitis (*n* = 106)	Osteomyelitis (*n* = 115)	*p* Value ^a^
Sex			
Female	39 (37%)	39 (34%)	
Male	67 (63%)	76 (66%)	0.65
Age (year)			
Mean ± SD	54 ± 21	60 ± 18	<0.05
Range	<1–100	15–98	
Infected sites			
head and neck	30 (28%)	5 (4%)	<0.0001
trunk	27 (25%)	4 (3%)	<0.0001
limbs	49 (46%)	106 (92%)	<0.0001
Underlying diseases			
CHC	14 (13%)	34 (30%)	<0.05
CKD	2 (2%)	15 (13%)	<0.05
DM	39 (37%)	50 (43%)	0.31
ESRD	9 (8%)	8 (7%)	0.67
Gout	7 (7%)	5 (4%)	0.46
Cancer	8 (8%)	14 (12%)	0.25
Hospitalization ^b^			
Yes	28 (26%)	56 (49%)	<0.001
No	78 (74%)	59 (51%)	
Surgery ^c^			
Yes	20 (19%)	47 (41%)	<0.001
No	86 (81%)	68 (59%)	

^a^ Categorical variables between the two groups were compared by chi-square test, whereas continuous variables were analyzed using the two-tailed *t* test. ^b^ Within the previous 6 months prior to the culture date. ^c^ Within the previous 6 months prior to the culture date. Abbreviation: MRSA, methicillin-resistant *Staphylococcus aureus*; CHC, Chronic hepatitis and cirrhosis; CKD, Chronic kidney disease; DM, Diabetes mellitus; ESRD, End-stage renal disease.

**Table 2 jcm-08-00816-t002:** Antibiotic resistance profiles of the isolated CL-MRSA and OM-MRSA strains.

Antibiotic	MIC ^a^ (µg/mL)	MRSA Isolates, No. (%)		*p* Value ^b^
		CL-MRSA (*n* = 106)	OM-MRSA (*n* = 115)	
Ciprofloxacin	(MIC ≥ 4)	37 (35%)	69 (60%)	<0.001
Fusidic acid	(MIC ≥ 1)	3 (3%)	25 (22%)	<0.0001
Gentamicin	(MIC ≥ 16)	24 (23%)	59 (51%)	<0.0001
Rifampicin	(MIC ≥ 4)	0	2 (2%)	0.17
TMP-SMX	(MIC ≥ 4)	7 (7%)	30 (26%)	<0.0001
Vancomycin	(MIC ≥ 2)	0	6 (5%)	<0.05

^a^ MIC values determined by E-test. ^b^
*p* value by chi-square test. Abbreviations: CL, cellulitis; OM, osteomyelitis; MRSA, methicillin-resistant *Staphylococcus aureus*; MIC, minimum inhibitory concentration; TMP-SMX, trimethoprim-sulfamethoxazole.

**Table 3 jcm-08-00816-t003:** Molecular genotyping of the isolated CL-MRSA and OM-MRSA strains.

	SCC*mec* Type, No. (%)			MLST, No. (%)			*spa* Type, No. (%)	
	CL-MRSA (*n* = 106)	OM-MRSA (*n* = 115)		CL-MRSA (*n* = 106)	OM-MRSA (*n* = 115)		CL-MRSA (*n* = 106)	OM-MRSA (*n* = 115)
II	–	3 (3%)	ST5	1 (1%)	4 (3%)	t002	1 (1%)	4 (3%)
III	6 (6%)	27 (24%)	ST7	–	1 (1%)	t008	19 (18%)	20 (17%)
IV	59 (56%)	55 (48%)	ST8	22 (21%)	24 (21%)	t015	–	2 (2%)
V	41 (39%)	30 (26%)	ST9	1 (1%)	2 (2%)	t019	9 (8%)	2 (2%)
			ST30	9 (8%)	2 (2%)	t026	4 (4%)	1 (1%)
			ST45	6 (6%)	7 (6%)	t034	3 (3%)	–
			ST59	54 (51%)	37 (32%)	t037	7 (7%)	29 (25%)
			ST188	2 (2%)	2 (2%)	t091	–	1 (1%)
			ST239	7 (7%)	28 (24%)	t189	2 (2%)	2 (2%)
			ST241	–	1 (1%)	t345	–	2 (2%)
			ST338	1 (1%)	–	t437	38 (36%)	27 (23%)
			ST398	1 (1%)	2 (2%)	t441	12 (11%)	9 (8%)
			ST508	–	2 (2%)	t571	–	2 (2%)
			ST573	–	3 (3%)	t574	–	1 (1%)
			ST1232	2 (2%)	–	t899	1 (1%)	2 (2%)
						t967	1 (1%)	2 (2%)
						t1081	2 (2%)	6 (5%)
						t1380	1 (1%)	–
						t3485	1 (1%)	1 (1%)
						t3515	1 (1%)	–
						t3525	–	1 (1%)
						t3527	1 (1%)	–
						t8391	–	1 (1%)
						new	3 (3%)	–

Abbreviations: CL, cellulitis; OM, osteomyelitis; MRSA, methicillin-resistant *Staphylococcus aureus*; SCC*mec*, staphylococcal cassette chromosome *mec*; MLST, multilocus sequence typing; ST, sequence type; *spa*, staphylococcal protein A.

**Table 4 jcm-08-00816-t004:** The relationship between the antibiotic resistance and MLST genotypes for CL-MRSA isolates (*n* = 106).

Antibiotic	MIC ^a^ (µg/mL)	CL-MRSA Isolates, %										
		ST5 (*n* = 1)	ST8 (*n* = 22)	ST9 (*n* = 1)	ST30 (*n* = 9)	ST45 (*n* = 6)	ST59 (*n* = 54)	ST188 (*n* = 2)	ST239 (*n* = 7)	ST338 (*n* = 1)	ST398 (*n* = 1)	ST1232 (*n* = 2)
Ciprofloxacin	(MIC ≥ 4)	100	95	0	11	0	9	100	100	0	0	0
Fusidic acid	(MIC ≥ 1)	0	5	0	11	0	0	0	14	0	0	0
Gentamicin	(MIC ≥ 16)	100	9	0	11	0	19	100	100	0	0	50
Rifampicin	(MIC ≥ 4)	0	0	0	0	0	0	0	0	0	0	0
TMP-SMX	(MIC ≥ 4)	0	0	0	0	0	2	0	86	0	0	0
Vancomycin	(MIC ≥ 2)	0	0	0	0	0	0	0	0	0	0	0

^a^ MIC values determined by E-test. Abbreviations: MLST, multilocus sequence typing; CL, cellulitis; MRSA, methicillin-resistant *Staphylococcus aureus*; MIC, minimum inhibitory concentration; ST, sequence type; TMP-SMX, trimethoprim-sulfamethoxazole.

**Table 5 jcm-08-00816-t005:** The relationship between the antibiotic resistance and MLST genotypes for OM-MRSA isolates (*n* = 115).

Antibiotic	MIC ^a^ µg/mL)	OM-MRSA Isolates, %												
		ST5 (*n* = 4)	ST7 (*n* = 1)	ST8 (*n* = 24)	ST9 (*n* = 2)	ST30 (*n* = 2)	ST45 (*n* = 7)	ST59 (*n* = 37)	ST188 (*n* = 2)	ST239 (*n* = 28)	ST241 (*n* = 1)	ST398 (*n* = 2)	ST508 (*n* = 2)	ST573 (*n* = 3)
Ciprofloxacin	(MIC ≥ 4)	75	0	83	100	0	100	11	100	100	100	50	0	33
Fusidic acid	(MIC ≥ 1)	0	100	21	50	0	29	16	0	32	0	0	0	33
Gentamicin	(MIC ≥ 16)	75	0	13	100	50	29	49	50	89	100	0	0	100
Rifampicin	(MIC ≥ 4)	50	0	0	0	0	0	0	0	0	0	0	0	0
TMP-SMX	(MIC ≥ 4)	0	0	0	0	0	0	0	50	100	100	0	0	0
Vancomycin	(MIC ≥ 2)	0	0	0	0	0	0	3	0	14	100	0	0	0

^a^ MIC values determined by E-test. Abbreviations: MLST, multilocus sequence typing; OM, osteomyelitis; MRSA, methicillin-resistant *Staphylococcus aureus*; MIC, minimum inhibitory concentration; ST, sequence type; TMP-SMX, trimethoprim-sulfamethoxazole.

**Table 6 jcm-08-00816-t006:** Association between the antibiotic resistance profiles and the genotypes for CL-MRSA isolates.

Antibiotic	MIC (µg/mL)	CL-MRSA Isolates, No. (%)										
		ST8 (*n* = 22)		ST45 (*n* = 6)		ST59 (*n* = 54)					ST239 (*n* = 7)	
		t008-IV (*n* = 18)	non-t008-IV ^a^ (*n* = 4)	t026-IV (*n* = 4)	t1081-IV(V) ^b^ (*n* = 2)	t437-IV (*n* = 10)	t437-V (*n* = 27)	t441-IV (*n* = 6)	t441-V (*n* = 6)	non-t437/non-t441 ^c^ (*n* = 5)	t037-III (*n* = 6)	non-t037-III ^d^ (*n* = 1)
Ciprofloxacin	(MIC ≥ 4)	18 (100%)	3 (75%)	0	0	0	3 (11%)	1 (17%)	0	1 (20%)	6 (100%)	1 (100%)
Fusidic acid	(MIC ≥ 1)	1 (6%)	0	0	0	0	0	0	0	0	0	1 (100%)
Gentamicin	(MIC ≥ 16)	1 (6%)	1 (25%)	0	0	0	5 (19%)	2 (33%)	0	3 (60%)	6 (100%)	1 (100%)
Rifampicin	(MIC ≥ 4)	0	0	0	0	0	0	0	0	0	0	0
TMP-SMX	(MIC ≥ 4)	0	0	0	0	0	1 (4%)	0	0	0	6 (100%)	0
Vancomycin	(MIC ≥ 2)	0	0	0	0	0	0	0	0	0	0	0

^a^ non-t008-IV: t008-V (*n* = 1), t967-IV (*n* = 1), and new-IV (*n* = 2). ^b^ t1081-IV (V): t1081-IV (*n* = 2). ^c^ non-t437/non-t441: t1380-V (*n* = 1), t3485-V (*n* = 1), t3515-IV (*n* = 1), t3527-IV (*n* = 1), and new-IV (*n* = 1). ^d^ non-t037-III: t037-IV (*n* = 1). Abbreviations: CL, cellulitis; MRSA, methicillin-resistant *Staphylococcus aureus*; MIC, minimum inhibitory concentration; ST, sequence type; TMP-SMX, trimethoprim-sulfamethoxazole.

**Table 7 jcm-08-00816-t007:** Association between the antibiotic resistance profiles and the genotypes for OM-MRSA isolates.

Antibiotic	MIC (µg/mL)	OM-MRSA Isolates, No. (%)										
		ST8 (*n* = 24)		ST45 (*n* = 7)		ST59 (*n* = 37)					ST239 (*n* = 28)	
		t008-IV (*n* = 18)	non-t008-IV ^a^ (*n* = 6)	t026-IV (*n* = 1)	t1081-IV(V) ^b^ (*n* = 6)	t437-IV (*n* = 15)	t437-V (*n* = 12)	t441-IV (*n* = 2)	t441-V (*n* = 6)	non-t437/non-t441 ^c^ (*n* = 2)	t037-III (*n* = 25)	non-t037-III ^d^ (*n* = 3)
Ciprofloxacin	(MIC ≥ 4)	16 (89%)	4 (67%)	1 (100%)	6 (100 %)	2 (13%)	1 (8%)	1 (50%)	0	0	25 (100%)	3 (100%)
Fusidic acid	(MIC ≥ 1)	5 (28%)	0	0	2 (33%)	1 (7%)	2 (17%)	0	2 (33%)	1 (50%)	6 (12%)	3 (100%)
Gentamicin	(MIC ≥ 16)	2 (11%)	1 (17%)	0	2 (33%)	9 (60%)	5 (42%)	0	4 (67%)	0	22 (88%)	3 (100%)
Rifampicin	(MIC ≥ 4)	0	0	0	0	0	0	0	0	0	0	0
TMP-SMX	(MIC ≥ 4)	0	0	0	0	0	0	0	0	0	25 (100%)	3 (100%)
Vancomycin	(MIC ≥ 2)	0	0	0	0	0	0	0	1 (17%)	0	3 (12%)	1 (33%)

^a^ non-t008-IV: t008-V (*n* = 2), t441-IV (*n* = 1), t574-IV (*n* = 1), and t967-IV (*n* = 2). ^b^ t1081-IV (V): t1081-IV (*n* = 1) and t1081-V (*n* = 5). ^c^ non-t437/non-t441: t3485-V (*n* = 1) and t8391-V (*n* = 1). ^d^ non-t037-III: t037-IV (*n* = 2) and t037-V (*n* = 1). Abbreviations: OM, osteomyelitis; MRSA, methicillin-resistant *Staphylococcus aureus*; MIC, minimum inhibitory concentration; ST, sequence type; TMP-SMX, trimethoprim-sulfamethoxazole.

**Table 8 jcm-08-00816-t008:** Numbers/intervals of operations and admissions in cellulitis and osteomyelitis.

MRSA Isolates	Numbers of Surgical Operations (Debridement)		Numbers of Hospital Admissions		Intervals between Operations (Month)		Intervals between Admissions (Month)	
	Cellulitis	Osteomyelitis	Cellulitis	Osteomyelitis	Cellulitis	Osteomyelitis	Cellulitis	Osteomyelitis
Total strains	1.4 ± 2.5 (*n* = 106)	4.2 ± 6.5 (*n* = 115)	1.5 ± 2.1 (*n* = 106)	3.2 ± 3.4 (*n* = 115)	0.6 ± 1.0 (*n* = 106)	0.7 ± 0.8 (*n* = 115)	0.5 ± 0.5 (*n* = 106)	0.4 ± 0.4 (*n* = 115)
ST8	1.7 ± 2.6 (*n* = 22)	4.9 ± 10.9 (*n* = 24)	2.1 ± 2.7 (*n* = 22)	3.3 ± 3.6 (*n* = 24)	0.6 ± 1.0 (*n* = 22)	0.8 ± 0.7 (*n* = 24)	0.4 ± 0.6 (*n* = 22)	0.4 ± 0.4 (*n* = 24)
ST59	1.1 ± 2.1 (*n* = 54)	3.3 ± 3.9 (*n* = 37)	1.4 ± 1.7 (*n* = 54)	2.6 ± 2.3 (*n* = 37)	0.7 ± 1.2 (*n* = 54)	0.7 ± 0.8 (*n* = 37)	0.7 ± 0.5 (*n* = 54)	0.5 ± 0.5 (*n* = 37)
ST239	5.1 ± 4.2 ^a, b^ (*n* = 7)	6.1 ± 5.6 (*n* = 28)	3.3 ± 4.2 (*n* = 7)	3.9 ± 3.4 (*n* = 28)	0.7 ± 1.4 (*n* = 7)	0.9 ± 1.1 (*n* = 28)	0.3 ± 0.3 (*n* = 7)	0.3 ± 0.3 (*n* = 28)

^a^ ANOVA: *p* < 0.05, ST239 vs. ST8 in cellulitis. ^b^ ANOVA: *p* < 0.001, ST239 vs. ST59 in cellulitis. Data was presented as means ± SD. Abbreviation: ST, sequence type.

**Table 9 jcm-08-00816-t009:** Effect of chronic liver/kidney diseases on the morbidity of osteomyelitis caused by different MRSA genotypes.

	No. (%)	Numbers of Surgical Operations (Debridement)	Numbers of Hospital Admissions	Intervals between Operations (Month)	Intervals between Admissions (Month)
ST8 (*n* = 24)					
Yes ^a^	9 (37.5%)	4.8 ± 5.7	3.7 ± 2.1	0.4 ± 0.5	0.3 ± 0.3
No ^b^	15 (62.5%)	4.9 ± 13.3	3.1 ± 4.3	1.0 ± 0.7	0.5 ± 0.5
ST59 (*n* = 37)					
Yes	11 (30%)	3.8 ± 3.2	2.4 ± 1.6	0.9 ± 0.7	0.5 ± 0.4
No	26 (70%)	3.1 ± 4.2	2.7 ± 2.5	0.6 ± 0.9	0.5 ± 0.5
ST239 (*n* = 28)					
Yes	21 (75%)	7.4 ± 5.9 ^c^	4.6 ± 3.6	1.1 ± 1.1	0.3 ± 0.3
No	7 (25%)	2.1 ± 1.9 ^c^	1.9 ± 1.2	0.4 ± 0.7	0.3 ± 0.3

^a^ Yes: With chronic liver/kidney diseases. ^b^ No: Without chronic liver/kidney diseases. ^c^ T test: *p* < 0.05, Yes vs. No. Data was presented as means ± SD. Abbreviation: ST, sequence type.

**Table 10 jcm-08-00816-t010:** Antibiotic resistance profiles of OM-MRSA isolates from PJIs and non-PJIs.

Antibiotic	MIC ^a^ (µg/mL)	OM-MRSA Isolates, No. (%)		*p* Value ^b^
		PJIs (*n* = 26)	non-PJIs (*n* = 89)	
Ciprofloxacin	(MIC ≥ 4)	16 (62%)	53 (60%)	0.86
Fusidic acid	(MIC ≥ 1)	5 (19%)	20 (23%)	0.73
Gentamicin	(MIC ≥ 16)	13 (50%)	46 (52%)	0.88
Rifampicin	(MIC ≥ 4)	1 (4%)	1 (1%)	0.35
TMP-SMX	(MIC ≥ 4)	9 (35%)	21 (24%)	0.26
Vancomycin	(MIC ≥ 2)	0	6 (7%)	0.17

^a^ MIC values determined by E-test. ^b^
*p* value by chi-square test. Abbreviations: OM, osteomyelitis; MRSA, methicillin-resistant *Staphylococcus aureus*; PJI, periprosthetic joint infections; MIC, minimum inhibitory concentration; TMP-SMX, trimethoprim-sulfamethoxazole.

**Table 11 jcm-08-00816-t011:** Molecular genotyping analysis of OM-MRSA isolates from PJIs and non-PJIs.

SCC*mec* Type, No. (%)			MLST, No. (%)			*spa* Type, No. (%)			PVL Positive, No. (%)	
	PJIs (*n* = 26)	non-PJIs (*n* = 89)		PJIs (*n* = 26)	non-PJIs (*n* = 89)		PJIs (*n* = 26)	non-PJIs (*n* = 89)	PJIs (*n* = 26)	non-PJIs (*n* = 89)
II	1 (4%)	1 (1%)	ST5	1 (4%)	3 (3%)	t002	1 (4%)	3 (3%)	16 (62%)	48 (54%)
III	10 (39%)	17 (19%)	ST7	1 (4%)	–	t008	5 (19%)	15 (17%)		
IV	8 (31%)	48 (54%)	ST8	6 (23%)	18 (20%)	t015	1 (4%)	1 (1%)		
V	7 (27%)	23 (26%)	ST9	–	2 (2%)	t019	–	2 (2%)		
			ST30	–	2 (2%)	t026	–	1 (1%)		
			ST45	–	7 (8%)	t037	9 (35%)	20 (23%)		
			ST59	8 (31%)	29 (33%)	t091	1 (4%)	–		
			ST188	–	2 (2%)	t189	–	2 (2%)		
			ST239	9 (35%)	19 (21%)	t345	–	2 (2%)		
			ST241	–	1 (1%)	t437	8 (31%)	19 (21%)		
			ST398	–	2 (2%)	t441	1 (4%)	8 (9%)		
			ST508	1 (4%)	1 (1%)	t571	–	2 (2%)		
			ST573	–	3 (3%)	t574	–	1 (1%)		
						t899	–	2 (2%)		
						t967	–	2 (2%)		
						t1081	–	6 (7%)		
						t3485	–	1 (1%)		
						t3525	–	1 (1%)		
						t8391	–	1 (1%)		

Abbreviations: PJI, periprosthetic joint infections; SCC*mec*, staphylococcal cassette chromosome *mec*; MLST, multilocus sequence typing; ST, sequence type; *spa*, staphylococcal protein A; PVL, Panton–Valentine leukocidin.

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
