# Peer review of "Comparison of Methicillin-Resistant Staphylococcus aureus Isolates from Cellulitis and from Osteomyelitis in a Taiwan Hospital, 2016–2018"

_jcm, 2019, doi:10.3390/jcm8060816_

Round 1
Reviewer 1 Report
GENERAL COMMENTS
MRSA infections are an important topic that is appropriate for the journal. I generally liked the paper; my criticisms are meant to make it even better. I believe that the present story could be told in a more interesting way. As currently presented it is largely archival. Some sections of the Results are so full of acronyms and numbers that few readers will want to wade through the paragraphs.
The English usage is quite good, although there are some words that are either wrong or suboptimal. I mentioned some, but not all, below. I noticed that nobody was acknowledged for critical comments on the manuscript. I strongly urge the authors to have experienced colleagues review this manuscript to make the story more interesting and to improve English word use.
SPECIFIC COMMENTS
Line 52. Increased resistance. This is vague. Resistance is a clinical term based on breakpoints. Increased resistance generally means increased prevalence of resistance. The present use is vague lab jargon. To be more precise use reduced susceptibility.
Line 59 delete been
Line 206 the word no is singular
Line 227 this sentence does not make sense
Line 233 How do your designations fit with the terms HA-MRSA and CA-MRSA? These all need to be tied together to make the present paper relevant to other work.
Line 246. It would be good in the first or second paragraphs of the Discussion to tie your work into other work in which HA and CA MRSA are compared. For example, there is an idea that during hospitalization of CA-MRSA, some virulence is lost as weak patients are attached and drug resistance markers are picked up. In searching for this material look for Bo Shopsin as a search term. My point is that the current presentation is largely archival and you need something to make it more interesting.
Line 259. Implicate is suboptimal here. What about indicate?
Paragraph ending at line 281. This is a superficial treatment that undermines author credibility. For example, I would not say that you can reduce the occurrence of antibiotic resistance by the correct use of therapy. As long as you are applying pressure you will increase the emergence of resistance. You might say that you will not make it worse as fast as if you use inappropriate agents. Regarding cipro, this derivative is not as good against Gram-positives as some other fluoroquinolones. HA-MRSA is widely cipro resistant. Rifampicin is widely known as quickly creating resistant S. aureus and is generally not used as monotherapy. I suggest that the authors become more familiar with the literature on antibiotic resistance.
Line 276 oral fluoroquinolone. Are there fluoroquinolones in use that are not oral? If all are oral, this statement makes the authors appear to be uninformed.
Author Response
Response to Reviewer 1 Comments
GENERAL COMMENTS
MRSA infections are an important topic that is appropriate for the journal. I generally liked the paper; my criticisms are meant to make it even better. I believe that the present story could be told in a more interesting way. As currently presented it is largely archival. Some sections of the Results are so full of acronyms and numbers that few readers will want to wade through the paragraphs.
The English usage is quite good, although there are some words that are either wrong or suboptimal. I mentioned some, but not all, below. I noticed that nobody was acknowledged for critical comments on the manuscript. I strongly urge the authors to have experienced colleagues review this manuscript to make the story more interesting and to improve English word use.
Response: We are grateful for the reviewer’s comments and positive evaluation of our work. We have now revised and modified the manuscript according to the reviewer’s comments. Additionally, the revised manuscript has been edited by Prof. Hsin-Nung Shih who is an expert in the field and also familiar with academic writing.
SPECIFIC COMMENTS
Point 1: Line 52. Increased resistance. This is vague. Resistance is a clinical term based on breakpoints. Increased resistance generally means increased prevalence of resistance. The present use is vague lab jargon. To be more precise use reduced susceptibility.
Response 1: As recommended by the reviewer, we have changed “increased resistance” to “reduced susceptibility” in the revised manuscript (Page 2; Line 54).
Point 2: Line 59 delete been
Response 2: The word “been” has now been deleted (Page 2; Lin 63).
Point 3: Line 206 the word no is singular
Response 3: The word “no” has now been revised to “few” (Page 16; Line 279).
Point 4: Line 227 this sentence does not make sense
Response 4: In the revised manuscript, this sentence has been revised (Page 18; Lines 314-315).
Point 5: Line 233 How do your designations fit with the terms HA-MRSA and CA-MRSA? These all need to be tied together to make the present paper relevant to other work.
Response 5: In the revised manuscript, we have now included additional information about CA-MRSA and OM-MRSA in the Introduction (Page 2; Lines 67-74) and Discussion (Page 18; Lines 315-319) sections. Due to the fact that several studies have shown that the distinctions between CA-MRSA and HA-MRSA have now become increasingly blurred and have little predictive value for clinical practice, we here did not collect the data in the study.
Point 6: Line 246. It would be good in the first or second paragraphs of the Discussion to tie your work into other work in which HA and CA MRSA are compared. For example, there is an idea that during hospitalization of CA-MRSA, some virulence is lost as weak patients are attached and drug resistance markers are picked up. In searching for this material look for Bo Shopsin as a search term. My point is that the current presentation is largely archival and you need something to make it more interesting.
Response 6: In the revised manuscript, we have provided additional information about the classifications of HA-MRSA and CA-MRSA in the first paragraph of the Discussion (Page 18; Lines 315-319). Furthermore, the emergence of multi-drug resistant MRSA lineages has been discussed on Page 19 (Lines 397-402) (Shopsin et al., 2019).
Point 7: Line 259. Implicate is suboptimal here. What about indicate?
Response 7: We have modified the paragraph in the revised manuscript (Page 18; Line 348-354) to try to point out the importance of our findings.
Point 8: Paragraph ending at line 281. This is a superficial treatment that undermines author credibility. For example, I would not say that you can reduce the occurrence of antibiotic resistance by the correct use of therapy. As long as you are applying pressure you will increase the emergence of resistance. You might say that you will not make it worse as fast as if you use inappropriate agents. Regarding cipro, this derivative is not as good against Gram-positives as some other fluoroquinolones. HA-MRSA is widely cipro resistant. Rifampicin is widely known as quickly creating resistant S. aureus and is generally not used as monotherapy. I suggest that the authors become more familiar with the literature on antibiotic resistance.
Response 8: We thank the reviewer for this comment. We have made the necessary changes in the paragraph in the revised manuscript (Pages 18-19; Lines 361-402). The emergence of multi-drug resistant MRSA lineages has also been discussed more fully in the paragraph (Page 19; Lines 397-402).
Point 9: Line 276 oral fluoroquinolone. Are there fluoroquinolones in use that are not oral? If all are oral, this statement makes the authors appear to be uninformed.
Response 9: Thank you for pointing this out. Correction has been made in the revised manuscript (Page 19; Lines 391-392).
Reviewer 2 ReportThe manuscript concerns isolates from cellulitis and osteomyelitis at a Taiwan hospital between 2016 and 2018. While the manuscript contains detailed data regarding the current epidemiology regarding skin and soft tissue infectious vs osteomyelitis isolates, it is too lengthy and has too many tables. This makes it difficult to follow. There are also some errors in the English Grammar/word choice and the manuscript should be reviewed for the English language. Detailed comments and suggestions can be found below:
1. Introduction. CA and HA are not definitions that relates to the isolates but to where the infections was acquired- in the community vs in the hospital. While it is true that CA-MRSA are most often SCCmec type IV, the same strains can also be transmitted within a hospital setting.
2. Methods. Since the authors are describing antimicrobial susceptibility patterns, further describing the patient population in terms of where the infection was acquired would have been informative, using already established guidelines. Furthermore, patient characteristics such as underlying conditions, previous hospitalizations, surgeries etc should be described.
3. Methods. For SCCmec, MLST, spa typing, if the already established primers and methods were used, there is not need for supplemental tables, only references. The same is true for the PVL genes PCR primers, which are both presented in the methods and in a supplemental table.
4. Results 3.1: years are omitted in the age section. Was it not expected to find the OM isolates most commonly derived from the limbs and the CL isolates derived from all parts of the body?
5. Table 2: This would be more interesting if the epidemiology (HA vs CA) was available for these patients to better understand previous exposures to the healthcare settings, etc.
6. 3.3- MLST and ST are already defined
7. The sentence beginning with “Specially” should be changed.
8. 3.3 -The isolates are PVL-genes positive.
9. 3.4 again, it would be of interest to know the epidemiology of these infections to understand if the difference in resistance between CL and OM isolates related to the source or the exposure.
10. Table 3. PVL positivity can be presented in text only and deleted from the table.
11. Table 5. Where the OM isolates isolated at early in the infection, prior to treatment? Is there only one isolate per patient? Did the patients have previous infections? Previous Staphylococcus aureus infections?
12. Tables 6 and 7. The numbers for some of these STs are very small and the tables could be supplemental or deleted.
13. 3.5 It can be expected that patients with OM undergo more surgeries. OM and CL should not be compared with regards to clinical management, such as in Table 8. Table 8 is difficult to understand. What kind of operations were performed for patients with cellulitis? Incision and drainage? Does this table represent recurrent infections? Were patients with cellulitis admitted to the hospital?
14. Table 9. This table is difficult to understand. Did, for example, patients with ST239 on average have 7.4 surgeries to treat their infection?
15. Tables 13 and 14 can be deleted.
16. Discussion: line 245-why did the authors expect an association between PVL genes presence and antimicrobial resistance?
17. Line 249. It should be expected to find more surgeries for OM vs CL. These should not be compared in terms of management.
18. Line 264. Again, since the ST is likely unknown to the treating physician, it would have been much more informative to know how the epidemiology related to the susceptibility patterns.
19. Supplements describing already published primers are unnecessary. These should be referenced in the methods section.
Author Response
Response to Reviewer 2 Comments
Comments and Suggestions for Authors:
The manuscript concerns isolates from cellulitis and osteomyelitis at a Taiwan hospital between 2016 and 2018. While the manuscript contains detailed data regarding the current epidemiology regarding skin and soft tissue infectious vs osteomyelitis isolates, it is too lengthy and has too many tables. This makes it difficult to follow. There are also some errors in the English Grammar/word choice and the manuscript should be reviewed for the English language. Detailed comments and suggestions can be found below:
Response: We thank this reviewer for the comments and suggestions. We have now revised and modified the manuscript according to the reviewer’s comments. Additionally, the revised manuscript has been edited by Prof. Hsin-Nung Shih who is an expert in the field and also familiar with academic writing. The response to each concern follows:
Point 1: Introduction. CA and HA are not definitions that relates to the isolates but to where the infections was acquired- in the community vs in the hospital. While it is true that CA-MRSA are most often SCCmec type IV, the same strains can also be transmitted within a hospital setting.
Response 1: In the revised manuscript, we have now added the definitions of CA-MRSA and HA-MRSA to the Introduction section (Page 2; Lines 56-59).
Point 2: Methods. Since the authors are describing antimicrobial susceptibility patterns, further describing the patient population in terms of where the infection was acquired would have been informative, using already established guidelines. Furthermore, patient characteristics such as underlying conditions, previous hospitalizations, surgeries etc should be described.
Response 2: We thank the reviewer for this suggestion. In the revised manuscript, we have modified the Methods section (Page 2; Lines 90-93), and have provided a revised Table 1 that includes analysis of patient characteristics such as underlying diseases and medical conditions.
Point 3: Methods. For SCCmec, MLST, spa typing, if the already established primers and methods were used, there is not need for supplemental tables, only references. The same is true for the PVL genes PCR primers, which are both presented in the methods and in a supplemental table.
Response 3: As recommended by the reviewer, these supplementary tables that are related to the already published PCR primers are now removed from the manuscript.
Point 4: Results 3.1: years are omitted in the age section. Was it not expected to find the OM isolates most commonly derived from the limbs and the CL isolates derived from all parts of the body?
Response 4: We apologize for the lack of clarity of the set of data. A revised Table 1 is now provided in the manuscript. Additionally, we agree with the reviewer that it was expected to find the OM isolates most commonly derived from limbs and the CL isolates derived from all parts of the body. We have made some changes in the text to reflect the expectation (Page 3; Line 155).
Point 5: Table 2: This would be more interesting if the epidemiology (HA vs CA) was available for these patients to better understand previous exposures to the healthcare settings, etc.
Response 5: We agree with the reviewer that the distinction between CA-MRSA and HA-MRSA would be important to better understand previous exposures to the healthcare settings. At the current stage, we think that the classification between CA-MRSA and HA-MRSA, along with the distinction between CL-MRSA and OM-MRSA, could be potentially interesting, but beyond the scope of the present report. In the revised manuscript, we have included additional information to the Introduction section (Page 2; Lines 67-74) and to the Discussion section (Page 18; Lines 315-319) to describe and discuss the issues about CA-MRSA and OM-MRSA.
Point 6: 3.3- MLST and ST are already defined
Response 6: The sentence has been corrected (Page 7; Lines 192-193).
Point 7: The sentence beginning with “Specially” should be changed.
Response 7: The word “Specially” has been changed to “Herein” in the sentence (Page 7; Line 201).
Point 8: 3.3 -The isolates are PVL-genes positive.
Response 8: As recommended by the reviewer, the sentence has been revised (Page 7; Lines 204-205).
Point 9: 3.4 again, it would be of interest to know the epidemiology of these infections to understand if the difference in resistance between CL and OM isolates related to the source or the exposure.
Response 9: We agree with the reviewer that the distinction between CA-MRSA and HA-MRSA would be important to better understand previous exposures to the healthcare settings. We think that the distinction between CA-MRSA and HA-MRSA in the study could be potentially interesting, but beyond the scope of the present report. In the revised manuscript, we have added some sentences in the Introduction section (Page 2; Lines 67-74) and in the Discussion section (Page 18; Lines 315-319) to describe and discuss the issues about CA-MRSA and OM-MRSA.
Point 10: Table 3. PVL positivity can be presented in text only and deleted from the table.
Response 10: As recommended by the reviewer, we have now deleted the PVL positivity in Table 3.
Point 11: Table 5. Where the OM isolates isolated at early in the infection, prior to treatment? Is there only one isolate per patient? Did the patients have previous infections? Previous Staphylococcus aureus infections?
Response 11: In the study, all OM or CL isolates were obtained from different patients (one isolate per patient). This point has been included in the Methods section (Page 2; Line 91). Additionally, patients from whom MRSA was isolated prior to the study period were excluded from the study.
Point 12: Tables 6 and 7. The numbers for some of these STs are very small and the tables could be supplemental or deleted.
Response 12: Since Tables 6 and 7 contain a wealth of information about the association between the antibiotic resistance profiles and MRSA genotypes, we would like to maintain these Tables in the main text, please.
Point 13: 3.5 It can be expected that patients with OM undergo more surgeries. OM and CL should not be compared with regards to clinical management, such as in Table 8. Table 8 is difficult to understand. What kind of operations were performed for patients with cellulitis? Incision and drainage? Does this table represent recurrent infections? Were patients with cellulitis admitted to the hospital?
Response 13: There are five specific criticisms embedded in the comment. The response to each criticism follows: (1) In the revised manuscript, we have now deleted the comparison between OM and CL with regard to clinical management (Page 13; Line 248); (2) We have modified the original Table 8 and try to make it clear (revised Table 8); (3) The surgical operation defined in the study is “radical debridement” that includes the removal of necrotic skin and subcutaneous soft tissue from wound, as well as wound irrigation with copious amounts of normal saline. This point has been added to Table 8; (4) Data from Table 8 do not represent recurrent infections. As some cases had massive necrotic soft tissue infection, debridement had to be done multiple times; (5) Yes, patients with severe cellulitis were admitted to the hospital (Table 8).
Point 14: Table 9. This table is difficult to understand. Did, for example, patients with ST239 on average have 7.4 surgeries to treat their infection?
Response 14: We have revised the original Table 9 and try to make it easy for readers to understand. Patients with acute osteomyelitis caused by ST239 could develop into persistent bacterial infection, debridement had to be performed multiple times until infection removed.
Point 15: Tables 13 and 14 can be deleted.
Response 15: Tables 12 and 13 have been moved to the supplementary data file.
Point 16: Discussion: line 245-why did the authors expect an association between PVL genes presence and antimicrobial resistance?
Response 16: Since earlier studies have reported that PVL genes are predominantly found in CA-MRSA, we therefore try to link an association between antimicrobial resistance and the presence of PVL genes in MRSA strains.
Point 17: Line 249. It should be expected to find more surgeries for OM vs CL. These should not be compared in terms of management.
Response 17: It is true that more surgeries would be expected for OM vs CL. We have now removed the comparison between OM and CL with regard to clinical management, and only emphasize that patients with cellulitis caused by ST239 need more surgeries as compared to patients caused by other MRSA genotypes (Page 18; Lines 340-343).
Point 18: Line 264. Again, since the ST is likely unknown to the treating physician, it would have been much more informative to know how the epidemiology related to the susceptibility patterns.
Response 18: We agree with the reviewer that the ST is likely unknown to the treating physician. We have now modified the statement in the revised manuscript (Page 18; Lines 359-360).
Point 19: Supplements describing already published primers are unnecessary. These should be referenced in the methods section.
Response 19: As recommended by the reviewer, supplementary tables that are related to the already published primers are now removed from the manuscript.